# Touchscreen Tasks for Cognitive Testing in Domestic Goats (*Capra hircus*): A Pilot Study Using Odd-Item Search Training

**DOI:** 10.3390/ani15142115

**Published:** 2025-07-17

**Authors:** Jie Gao, Yumi Yamanashi, Masayuki Tanaka

**Affiliations:** 1The Hakubi Center for Advanced Research and Wildlife Research Center, Kyoto University, 2-24 Tanaka-Sekiden-cho, Sakyo, Kyoto 606-8203, Japan; 2Center for Research and Education of Wildlife, Kyoto City Zoo, Kyoto 606-8333, Japan

**Keywords:** comparative cognition, domestic goats, touchscreen, odd-item search

## Abstract

The use of touchscreens for cognitive tests in animals has many advantages. In this study, a step-by-step method was used to train naïve goats to use a touchscreen successfully. The subsequent odd-item search tasks confirmed that they were able to perform cognitive tests using the touchscreen. The results from these tasks also suggested that the goats may have relied on a perceptual strategy rather than acquiring the rule of selecting the odd item.

## 1. Introduction

Comparative studies of animal cognition contribute to our understanding of how animals perceive and interact with their environment, how they use information to perform daily activities and solve problems, and how cognition has evolved. While this field has a long history of focusing on animals such as primates, birds, and rodents [1], cognitive studies in large farm animals have been developed over recent years [2,3,4]. Many farm animals are ungulates. Ungulates, as land mammals, are an important group in the course of evolution. Studying their cognition will help elucidate how cognitive capabilities evolve and change when mammals live on land. From the perspective of research regarding the evolution of cognition, studying such animal species will deepen our understanding of the effects of domestication on shaping the brain and perception. Farm animals have a close relationship with human society. From the perspectives of animal management and animal welfare, studying their cognition will reveal more information about what kinds of living conditions and environments could increase their well-being. Therefore, it is important to examine the cognitive abilities of these species.

Apart from using more traditional approaches, including observation and behavioral experiments [5,6,7,8,9,10], screen-based cognitive tests have been used in a series of domestic goat studies to examine their visual discrimination and learning [4,11,12,13,14,15,16]. In these studies, researchers set up a chamber with an LCD screen at one end. Four switches were located around the four corners of the screen. Four stimuli were shown on the screen, and they could be “chosen” by pushing the corresponding switches. This setup was used for visual discrimination and learning studies in domestic goats. The goats successfully discriminated many different shapes, revealing their high level of visual acuity, and validating test paradigms using stimuli on a screen [4,13,14,16].

Using a screen in cognitive tests has many advantages compared to using only physical objects, including paper or boards with printed images. First, it is much easier to present multiple stimuli. Second, all stimuli can be shown in a controlled manner. For example, actual objects and those on printed papers may appear slightly different from one another, but stimuli on a screen are of the same size and color. Third, if the test program is automated, the animals are prevented from using any other cues during the experiment, such as certain behaviors that human experimenters accidentally display when manipulating the stimuli and observing the animals’ responses. Fourth, if the test program is automated, results such as the response time can be recorded more accurately than by using a video recorder and coding the animals’ gaze direction. Fifth, in certain test contexts, using a screen to reveal multiple stimuli and provide rewards constitutes a universal reward system. For example, in a classic quantity discrimination task, the experimenter could show the animal two sets of food items with different numbers. In different trials, the numbers of food items could also differ, as could the contrast between the two choices in each trial. These differences in the amount of food rewards could potentially affect the animals’ performance. In such cases, the animals’ choices may not reflect their true ability for quantity discrimination, as the reward amount differs. If this experiment is conducted on a screen using 2D images as stimuli and a universal reward is provided for each correct choice regardless of the contrast of the two items for discrimination, the result would more accurately reflect their ability for quantity discrimination. Here, using physical picture presentations could also be effective, but using 2D presentations, combined with the features of having strictly controlled stimuli and procedures, forms an advantage of using a screen.

In the above-mentioned screen-testing device for goats, stimulus choice was achieved by pushing the switches. However, the association of pushing behavior and choosing a stimulus may take time for animals to learn and understand. In this setting, the presentation of stimuli was also constrained by the physical switches. To address these issues, a touchscreen system can be used instead. The use of touchscreens in cognitive tests is not uncommon in many species, including primates, rodents, and birds (e.g., [17,18,19,20,21,22,23,24]). Without the constraints of physical switches, stimuli could be presented in many ways, making it possible to have flexible experimental designs. Once they learned to use the touchscreen, the animals could quickly become experts at it because the operation is relatively intuitive. However, it can be difficult for ungulates to use the touchscreens and to apply this to cognitive research. Primates and mice use their hands in their daily activities, so they can learn to touch the screen naturally with their hands. Birds have pecking behaviors, so it could be intuitive for them to peck stimuli on a screen. However, ungulates do not have such appendages available to manipulate a touchscreen with, making their potential method of interaction with a touchscreen system less obvious.

Researchers have trained goats to use touchscreens in a similar setting as the previous non-touchscreen screens with switches, following a long history of research at the same facility [25,26]. A testing device was set up in the goats’ living area, and the goats obtained water by completing tasks. However, little research has been conducted in other settings where goats use touchscreens for cognitive tests. It can be challenging to train completely naïve animals for many reasons, including fear of the touchscreen (a novel object) and setup, the unfamiliarity of a touchscreen, and the unfamiliar process of completing a cognitive test. Several studies using touchscreens have been conducted in horses [27,28,29,30,31]. Similar methods were adopted in this study following the detailed description from two studies among them [30,31]. In both studies, the screens were divided into left and right parts by a physical frame. When the horses touched any point on either part of the screen, the stimulus on that part was registered as “chosen”. This method provides a template for training naïve ungulate animals to use a touchscreen; however, using only two stimuli on a touchscreen may be an experimental constraint because the number of stimuli is limited. If the animals could use the touchscreen freely, as other animals and humans do, various cognitive tests would be possible. The odd-item search is a visual search task in which the participant is meant to choose the odd-item against multiple distractors [32,33]. Previous studies in pigeons and chimpanzees reported better performance as the number of distractors was increased, suggesting a perceptual strategy instead of immediate learning of the rule to search for the odd-item [19,21,23,32]. One study showed that some exceptional goats could transfer the odd-item search rule to novel stimuli after extensive training [11]. In this study, odd-item search tasks were used. The goals are to confirm that the goats could complete the cognitive test using a touchscreen and to examine if the goats could transfer this rule by using multiple sets of stimuli. First, as a pilot test, one set of stimuli was used to test their performances when the number of distractors differed. Then, they were trained to learn this rule and were tested using multiple sets of novel stimuli in order to examine if they could transfer the rule successfully.

## 2. Methods

### 2.1. Participants

Four domestic goats (*Capra hircus*, Table 1) participated in this study. These goats live in a social group consisting of seven goats and one sheep at Kyoto City Zoo. Their outdoor compound is about 400 m^2^, enriched with climbing blocks, brushes for rubbing, sheds, and trees. There is a hallway outside the compound for zoo staff and experimenters, but not for zoo visitors. Several 3.9–7.9-m^2^ indoor rooms for the animals are across the hallway. The animals are in the outdoor compound between 09:00 and 16:00 but use the indoor compound when it is raining. The compound is open for zoo visitors twice daily, and visitors can interact with the animals. School children occasionally enter the grounds for educational programs. The experiments were conducted at noon when no visitors were allowed inside the compound. Water was available to the goats at all times. There was no food deprivation. This research was approved by the Ethical Committee at Kyoto City Zoo (#2023-KCZ-022, 2024-KCZ-002) and followed the animal welfare guidelines of the Kyoto City Zoo.

### 2.2. Device and Setting

A 23.8-inch touchscreen (Dell P2424HT, 1920 × 1080 pixels) set on a stand on movable wheels (Hayami TF-520B, Hayami Industry Co., Ltd., Nagahama, Japan) was used for the experiment. The center of the screen was about 105 cm from the ground, which is the same height as the heads of the goats when they stand near the screen. One goat, Ryu, is bigger than the others, so four small flowerpots were positioned under the four wheels to make the screen higher when he was being tested to make sure he could operate the touchscreen in a similar manner as the other goats.

The experiments were conducted in the hallway between the outdoor and indoor compounds: the computer, touchscreen, and leading experimenters all remained in the hallway. The goats mostly stayed inside the compound. On rare occasions, they were in an indoor room. One experimenter usually remained inside the goat compound to separate the goats into two groups and/or distract the goats when complete separation was not possible to make sure only one goat was near the touchscreen. The tested goat had visual and auditory access to the other members and could move freely. A horizontal wooden bar fence was situated between the compound and the hallway. The goats could stick their head and neck out through the open spaces in the fence without any difficulty. The height of the screen was adjusted so that the goats’ front hooves were in contact with the lower fence bar, and their hind hooves were in contact with the ground. This allowed the goats to maintain a natural position during a session. The experimenter was occasionally close to the goats to guide their touching behavior in the training stage but always stood behind the touchscreen facing one side of the goats during the follow-up tests and made no eye contact with them during the session. There was a space at the center of the touchscreen stand, immediately under the center of the screen, through which the experimenter gave food rewards to the goats. A fixed plate was initially set around that area as a feeder, as in horse studies [30,31], but the goats continued to chew the plate and were distracted from the experiments, so it was decided that experimenters should feed them by hand, appearing only when there was a reward.

### 2.3. Touchscreen Training

A round, black plastic plate stuck on one end of a long wooden stick was used to teach the goats to touch (Figure 1A). In the beginning, the experimenter placed the plate against the goats’ mouths, said “touch”, and gave them a piece of carrot as a reward. Later, the goats learned to touch the plate with their mouths or tongues upon hearing the instruction “touch” and received the reward.

Then, the equipment was changed to a whiteboard and magnetic sheets (Figure 1B). The experimenter showed the whiteboard to the goats and placed a black magnetic sheet on it. The behavior of touching the sheet was guided by the instruction “touch” and rewarded with a piece of carrot. The location of the sheet changed with every trial. The goats learned to touch the sheets.

Then, a touchscreen was used (Figure 1C). Each session consisted of 12 trials. For each trial, the food reward was a piece of carrot. All experimental programs were written and executed in Microsoft Visual Studio 2022. The touchscreen for the touch training had six virtual locations for the stimuli (three in the upper half of the screen and three in the lower half). The stimuli were black circles with a diameter of 400 pixels. A “beep” (fundamental frequency = 546 Hz) sounded at the beginning of each session, and a stimulus appeared on the screen. After the goat touched the target, a chime (fundamental frequency = 1063 Hz) sounded, and a food reward was given by the experimenter. First, the target appeared in one of the three locations in the upper or lower part of the screen (“one row”). Within each session, the targets all appeared in the upper or lower half of the screen. The touching behavior was guided and rewarded. If the goats were reluctant to touch the screen, the experimenter directed their attention to the target by presenting the food reward near the target location, tapping the back of the screen at the corresponding spot, or standing near the target when it appeared on one side of the screen. After the goats could touch the targets correctly without hesitation by themselves, they began the next condition (“two rows”), where the stimulus appeared in one of the six locations on the screen (either in the upper or lower part of the screen) during each trial of one session. After the goats could touch the targets correctly without hesitation, they underwent the subsequent odd-item search tasks.

### 2.4. Pilot Test

In the pilot test, the target was always a black circle, the same as that the goats had been trained to touch during the touchscreen training period. The distractor was a black triangle of a similar size to the target (Figure 2A). The number of distractors varied from 1 to 5, resulting in five conditions. The goats were tested in ascending order according to the number of distractors, from 1 to 5.

Each condition consisted of four sessions, and each session consisted of 12 trials, of which six were test trials with both the target and distractor(s), and six were warm-up trials, in which only the target was shown without any distractors (Appendix A). All sessions began with a warm-up trial.

A warm-up trial was as follows. Each trial began with a “beep” sound, and then one target appeared in one of the six virtual places on the screen. If the goats touched the target, a chime sounded, and a food reward, a piece of carrot, was given by the experimenter. A test trial also started with a “beep” sound, and then one target as well as distractors appeared on the screen, each stimulus occupying one of the six virtual sections. If the goats touched the target, a chime sounded, and a food reward was given by the experimenter. If the goats touched a distractor, a “Booo” (fundamental frequency = 2209 Hz) sounded, and instead of a food reward, a timeout of 3 s was imposed. After the goats made their choice, all stimuli disappeared. The inter-trial interval was 5 s.

The goats first received one warm-up session (for 12 trials, only target) on each testing day, and then they received approximately four test sessions. The locations of the target were counterbalanced, and the locations of the distractors were randomized. The sequences of the locations of each distractor in each trial were prepared beforehand. They were generated by a randomizing program and then were adjusted manually to avoid being the same as the target and other distractors, if there were any.

### 2.5. Odd-Item Search Training

For the above-mentioned pilot test, Nishiki completed it much later than the others. Therefore, the performances of the other three participants were examined, and the distractor number of the subsequent tests was decided based on their performances. For the other three participants, the performance/chance-level ratio was highest when there were five distractors. The subsequent tasks then used one target and five distractors in each trial.

As in the pilot test, each session consisted of 12 trials, and the food reward for correct choices was a piece of carrot. However, different from the pilot tests, all trials in the test sessions here did not consist of warm-up trials (target only). In each test session, each trial had one target and five distractors. There were three stages, and each stage used two different sets of stimuli (Figure 2B–D), which were made from default shapes and icons in Microsoft Powerpoint and Pixelmator. In each test session, six trials used one set of the stimuli, and the other six used the other set of stimuli. The order of the trials was randomized. The flow of each test trial was the same as described above for the pilot test. The stages also differed in whether or not warm-up sessions (consisting only of target-only trials) were given.

The criterion to pass each stage was having at most 1 error (out of 12 trials in total for one session) for two consecutive sessions. In Stage 1, if the criterion was not reached by trial 20, a correction procedure was introduced until the goats reached the criterion; then, the correction procedure was discontinued, and training continued until the goats reached the criterion. In the correction procedure, after each wrong choice, this trial was repeated until they made a correct choice, and the repetition was at most three times. Yoshino did not perform well initially nor respond well to the correction procedure, so she was excluded from this experiment. The experiment ended after the goats either reached the criterion (and moved to the next stage) or the 20th session during Stages 2 and 3.

In Stage 1, before all tests, the goats first received 8 warm-up sessions, which only had one target in each trial. The two targets of the two sets of stimuli (Figure 2B) appeared in random order, but for six trials each in each session (Appendix A). Then, on each testing day, they received one warm-up session first before receiving the test sessions. In Stage 2, they also received 8 warm-up sessions before the tests, but after these 8 sessions, they no longer received any other warm-up sessions. Each testing day started directly with testing sessions. In Stage 3, there were no warm-up sessions at all, and they directly received testing sessions. The locations of the targets were counterbalanced.

### 2.6. Data Analyses

For the pilot test and the subsequent odd-item search training, the goats’ choices were recorded in each trial. Binomial tests were conducted to compare their performances with the chance levels.

Regarding the pilot test, in each condition and for each individual, the number of correct choices in total was compared to the chance level using binomial tests. The chance levels were 50%, 33.33%, 25%, 20%, and 16.67% when the distractor number was 1, 2, 3, 4, and 5, respectively.

Regarding the subsequent odd-item search training, because the distractor number was always 5, the chance level was 16.67%. Binomial tests were conducted to compare performance with the chance level during each stage for each individual, including the overall performance across all sessions (for Stage 1, data from sessions of correction procedures were excluded), the performance during the first four sessions of each stage, and the performance during the last four sessions of each stage.

## 3. Results

### 3.1. Touchscreen Training

The participants, Hikari, Yoshino, Ryu, and Nishiki, completed 4, 5, 20, and 8 sessions, respectively (12 trials per session), for the “one-row” training. On average, they completed 111 ± 44 trials (mean ± SEM). For the “two-row” training, Hikari, Yoshino, Ryu, and Nishiki completed 10, 10, 7, and 17 sessions, respectively. On average, they underwent 132 ± 25 trials (mean ± SEM).

### 3.2. Pilot Test

The number of correct choices, accuracy, and binomial test results compared to the chance levels are shown in Figure 3 and Table 2. The performances were significantly better than the chance level under certain conditions for all individuals (Table 2).

### 3.3. Odd-Item Search Training

The performance data are shown in Figure 4 and Table 3. In Stage 1, Hikari completed 35 sessions, including 2 sessions of the correction procedure; Ryu completed 51 sessions, including 19 sessions of the correction procedure; and Yoshino completed 27 sessions, including 7 sessions of the correction procedure. Hikari completed 21 sessions, and Ryu completed 20 sessions during Stage 2. Hikari completed 20 sessions during Stage 3.

Yoshino did not respond well to the training or the correction procedure during Stage 1, so she was eliminated from the test. In Stage 2, Ryu could not reach the criterion after 20 sessions of training. Hikari made one error in session 20 and could potentially reach the criterion if having one or no errors in the next session, so she completed another session, reached the criterion, and moved to Stage 3. Hikari did not reach the criterion after 20 sessions of training during Stage 3. Significantly better-than-chance-level performance, for the last four sessions and overall, was observed for Hikari during stages 1 and 2, and for Ryu during Stage 1, although their performances were not better than the chance level during the first four sessions (Table 3). The data are also available in Appendix A.

## 4. Discussion

Four naïve domestic goats were successfully trained to use a touchscreen in cognitive tests. A series of follow-up odd-item search experiments confirmed that the goats could smoothly complete the cognitive tests using the touchscreen by themselves. Thus, the procedures used for touchscreen training were valid. The goats also exhibited consistent results in the odd-item search task in a previous study in goats [11]: they generally had better-than-chance-level performance under most conditions, albeit there were individual differences, and only exceptional individuals made the transfer to novel stimuli after extensive training.

The goats were trained using a touchscreen and showed the possible applications for future research. The use of screens during cognitive testing has several advantages. The stimuli and presentation thereof can be well controlled. The time needed to make and display the stimuli can be reduced, compared to doing so manually using objects. The experimental program allowed accurate recording of results. Moreover, because the experiment was well controlled and did not require much time between trials, the experiments were conducted efficiently, which created a more stable testing environment for the animals, possibly allowing them to focus more on the tasks and achieve more accurate results. The use of touchscreens is advantageous as their operation is intuitive; once they have learned, the animals do not forget how to use the touchscreens. The system introduced in this study is easy to set up and could be used in different locations and conditions. It was particularly helpful to train completely naïve animals who had never been exposed to cognitive tests or devices. Naïve animals may respond better to a step-by-step training method, as they may fear strange objects such as a touchscreen [34]. At the study site, the touchscreen was exposed to all goats before all the training, and some goats never approached it, while others approached it but were reluctant to touch the screen. Therefore, a gradual shaping [35] may be helpful in training naïve animals.

In two previous horse studies, a physical framework was used to separate the screen into two parts, left and right, to make the operation for the animals easier [30,31]. While this could be a good idea in some species, it may not be necessary once the animals learn how to use the touchscreen. In our follow-up test, the goats could accurately touch one target and avoid several distractors that were all shown on the screen. This allows more diverse experiment paradigms that may require more than two locations on the screen.

Limitations of the experimental methods could explain why there was a lack of studies on large farm animals and/or ungulates. Many tests that used to be difficult to conduct are now possible with the help of touchscreens. Many animal facilities, such as zoos, have been adopting touchscreen systems for animal research and welfare [17,20,36,37,38,39]. Performing tasks for food rewards could be a way to achieve cognitive enrichment [40,41,42]. Animal facilities could also use such tasks to test food preferences, thereby improving the living environment and well-being of the animals. Our method worked in domestic goats, and it could also provide a platform for establishing methods in other species. We hope this study will inspire future use of touchscreens in cognitive studies of goats and other species.

In the pilot test, Hikari, Ryu, and Yoshino all had better-than-chance-level performance when there were more than two distractors, and their performance did not decrease much as the number of distractors increased. This was consistent with findings from previous studies in other species, such as pigeons and chimpanzees, where performance was better as the number of distractors increased [19,21,23,32]. Similar to other species, this could be due to the effect of comparison, which made the odd items “pop up”. Their performances were poorer when there were five distractors. Additionally, Nishiki’s performance level decreased as the number of distractors increased, which could be because it became more difficult for them to process as the information amount increased. Nishiki was also the eldest of the four participants. Her age could be related to the tendency to fail as the number of elements on the screen increased. However, with a small sample size, it cannot be confirmed that this was the reason. The changes in performance with varying numbers of distractors as well as Nishiki’s decreased performance both suggested the use of a perceptual strategy rather than an immediate understanding of the odd-item search rule. However, they performed above chance levels in many conditions (4, 3, 4, and 4 conditions for Hikari, Yoshino, Ryu, and Nishiki, respectively, Table 2), suggesting learning within the setting and potential to further learn the rule.

Because Nishiki joined the project later than the other goats, we used data from the remaining three goats during the pilot test to determine the number of distractors in the subsequent odd-item search training. Although the performance of these three goats dropped when there were five distractors, the performance/chance-level ratio was still highest when there were five distractors. Therefore, we chose this number of distractors for the subsequent tasks using novel stimuli.

In these tasks with novel stimuli, the participants showed mixed performances and individual differences. A clear learning effect was observed during Stage 1 (Hikari and Ryu) and Stage 2 (Hikari). Their performances were not significantly better than chance when they started (the first four sessions), but they were significantly better than the chance level at the end of the stage (the last four sessions), and all had better-than-chance-level performance overall (Table 3). This result shows that the goats were able to learn how to complete the task with familiar stimuli, which is consistent with a previous study [11]. However, without evidence of successful transfer, this learning effect could reflect associative learning rather than rule learning.

There were individual differences and difficulties in transferring to novel stimuli. While Yoshino did not respond well to the corrections and training during Stage 1, during the later training stages, Ryu and Hikari were highly accurate. Ryu did not transfer the rule during Stage 2, but Hikari was able to do so after training. Yoshino’s performances were significantly worse than the chance level (Table 3), suggesting that she might have been avoiding the odd item. This could be from fear of novel items [34]. However, the goats received warm-up sessions of targets only before test sessions in this stage. It is possible that the contrast between the five distractors and one target made the target more “popping-up”, making it a “novel” item. This individual difference was consistent with the previous study in goats showing that their performances improved after training on hundreds of trials and that only exceptional goats made a successful transfer [11]. Failure to transfer was shown by their performance at the beginning of Stages 2 and 3 (Table 3). For Hiraki in Stage 3, she eventually failed to perform accurately after 20 training sessions. This difficulty in learning and transfer indicates that goats tend to use perceptual cues, such as remembering the combination of the target and distractor (associative learning), instead of using the odd-item search rule, which is consistent with previous studies in many species [19,21,23,32]. The significantly worse-than-chance-level performance of Ryu during Stage 2 and of Yoshino during Stage 1 also suggested failure to learn this rule.

The goats received different numbers of sessions during the touch training and Stage 1 of the odd-item search training. These differences might be expected to influence their performance; however, the data suggest that the number of training sessions was not directly related to individual outcomes. For instance, Hikari received the fewest touch training sessions (14) yet showed the best performance in both the pilot test and subsequent stages. Yoshino, who received a similar number of sessions (15), performed worse than Hikari, particularly in the later stages. In contrast, Ryu underwent the most sessions (27), and Nishiki received a comparable amount (25), but their performances in the pilot test differed. Notably, Ryu outperformed Yoshino in the later stages. In Stage 1, the number of sessions required to reach the criterion also varied across individuals. Yet again, these differences did not appear to be associated with performance outcomes. Although Ryu required the most sessions, he eventually met the criterion and proceeded to Stage 2, whereas Yoshino failed to improve even after correction procedures were introduced. It is possible that Ryu’s extensive training in Stage 1 may have reinforced pattern learning and hindered rule learning, particularly in comparison to Hikari. However, such an interpretation cannot be confirmed given the limited sample size. Future research should take this possibility into account, alongside consideration of individual differences. 

There were some limitations in this study. Regarding the pilot test, because it was conducted in order of increasing distractor numbers, there was the possibility of a learning effect. However, given the fact that the subjects had been exposed to the target (a black circle) from the beginning of touchscreen training, they were already very familiar with the target but not the distractor during the beginning of testing. Furthermore, because performance decreased near the end when the distractor number was 5 (particularly for Nishiki, whose performance was decreasing throughout the experiment), there may have been no learning effect in this experiment. Regardless, a better method would have been to randomize the order of the conditions. In the later tasks using novel stimuli, the use of correction procedures in Stage 1 could potentially reinforce the pattern learning rather than rule learning in the goats. Stage 1 testing was more difficult than the pilot tests, not only because the stimuli were novel, but also because there were no warm-up trials to remind the goats of the targets during the tests. The implementation of correction procedures was intended to increase their accuracy first. Nevertheless, the use of correction procedures could be avoided or shortened, especially if the testing situation allows extensive training for the animals.

## 5. Conclusions

In summary, four domestic goats were trained to use a touchscreen, and they were able to complete a series of cognitive tests using the touchscreen without any problems. Our results show that it might be possible for goats to learn odd-item search after extensive training. It is hoped that our methods and results will be helpful for future research and animal welfare practices in domestic goats and other species.

## Figures and Tables

**Figure 1 animals-15-02115-f001:**
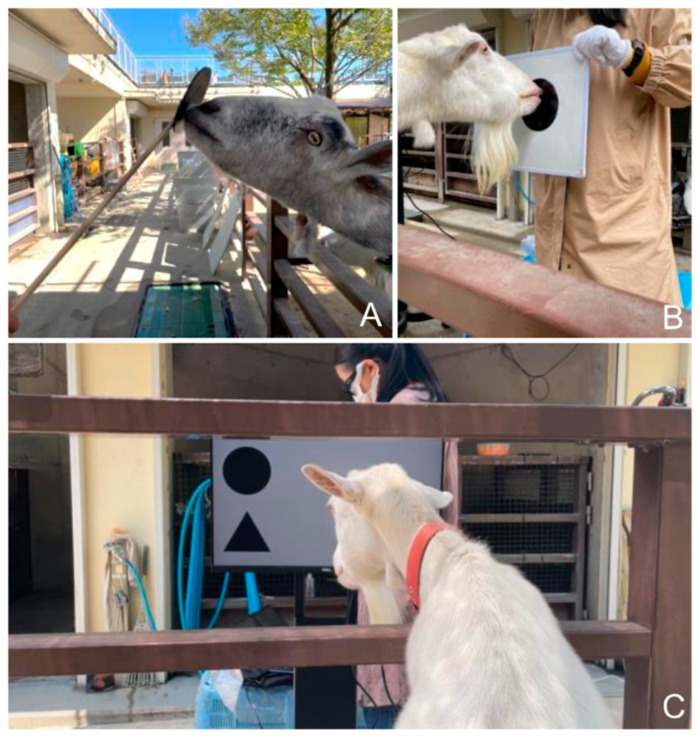
Touchscreen training and experimental setting. (**A**) Training the goats to touch a plastic plate. (**B**) Using a whiteboard and training the goats to touch a magnetic sheet. (**C**) The experimental setting. The leading human experimenter and the devices were in the hallway, while the goats remained in their living compound. They could easily stick their heads and necks out to reach the touchscreen and to perform the experiments. The experimenter stood behind the touchscreen, faced one side, and avoided visual contact with the goat during the experiment. Upon hearing the chime sound indicating a correct choice, the experimenter delivered a piece of carrot as a food reward to the goat through the space immediately under the touchscreen in the middle of the stand.

**Figure 2 animals-15-02115-f002:**
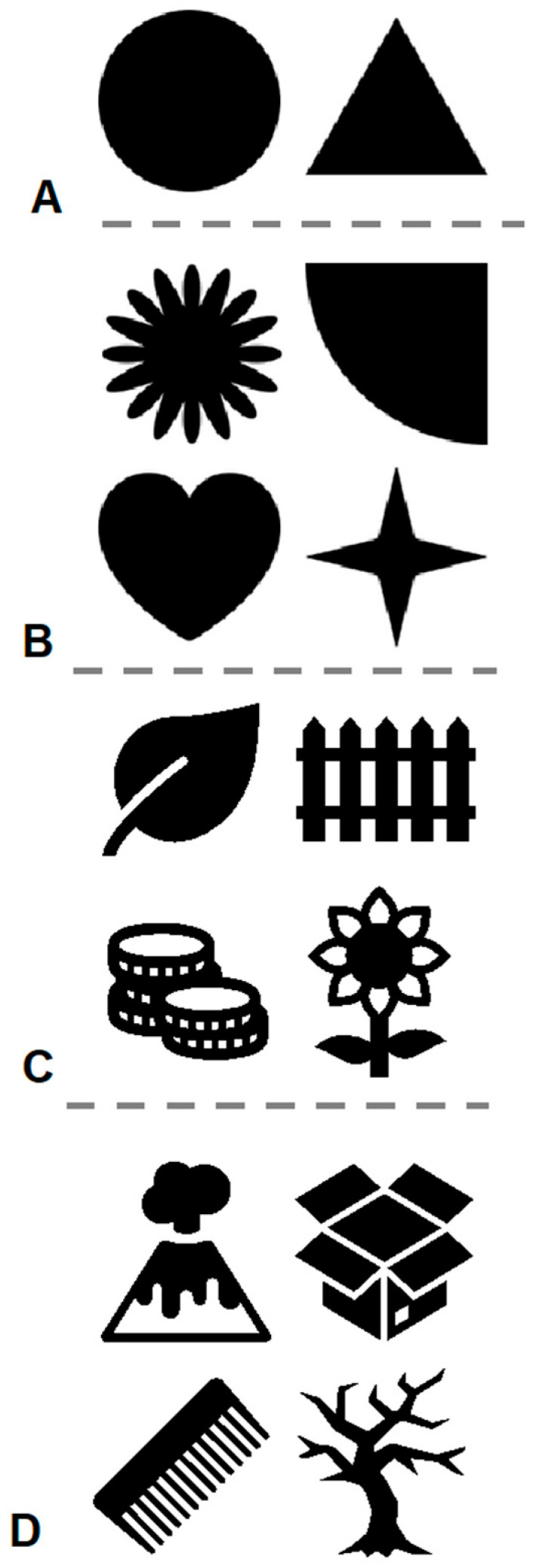
Stimuli used during the tests. (**A**) The stimuli used during the pilot test. The circle was the target, and the triangle was the distractor for all individuals. (**B**) The stimuli (two pairs; each row is one pair) used in odd-item training, Stage 1. For Hikari and Ryu, the left stimuli were the targets, and the right stimuli were the distractors. This was reversed for Yoshino. (**C**) The two pairs of stimuli used during Stage 2. The left stimuli were the targets, and the right stimuli were the distractors. (**D**) The two pairs of stimuli used during Stage 3. The left stimuli were the targets, and the right stimuli were the distractors.

**Figure 3 animals-15-02115-f003:**
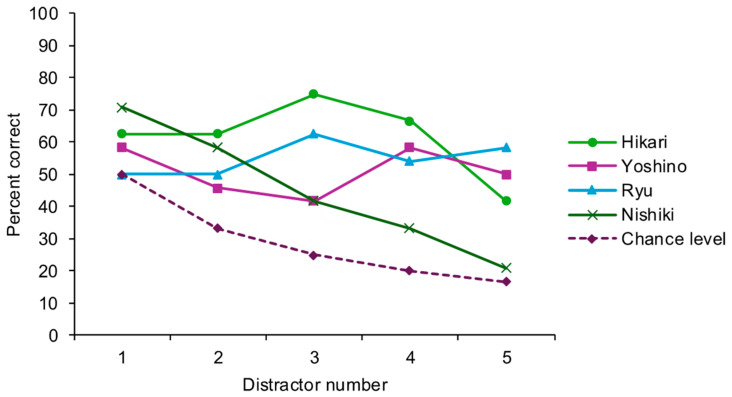
Performance of the pilot test. The figure shows the percent correct for each individual under each condition (the number of distractors ranged from one to five), accompanied by a dotted line showing the chance level for each condition.

**Figure 4 animals-15-02115-f004:**
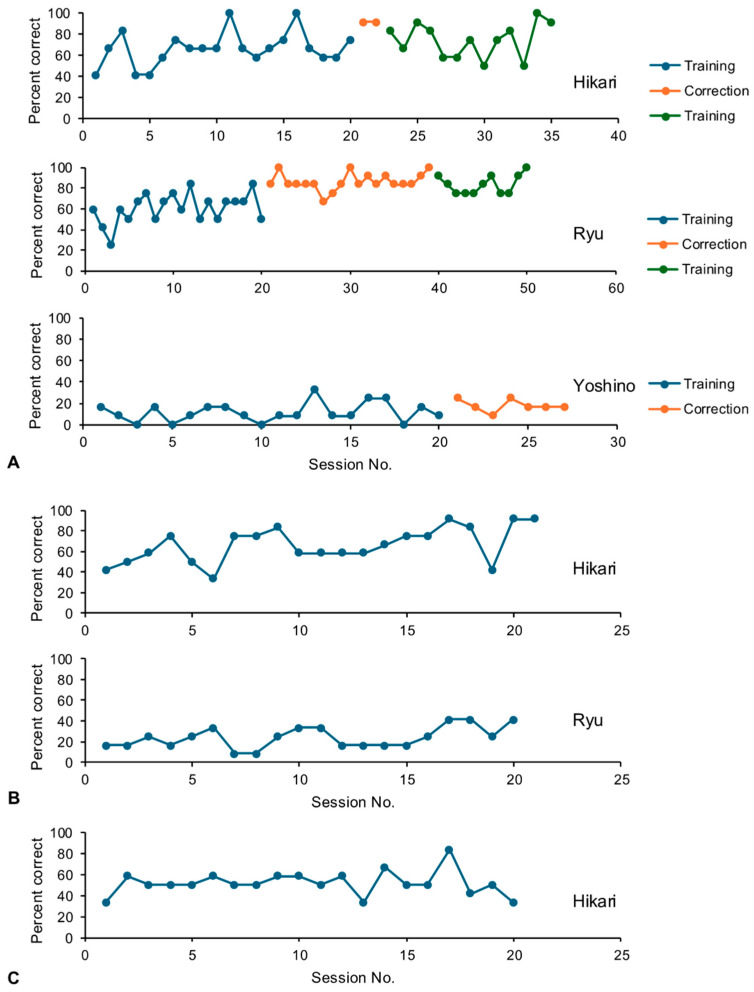
The learning curves for each individual at each stage of the odd-item search training performed after the pilot test. The names in the bottom right of each sub-figure are those of the participants. (**A**) Stage 1 performances. Correction procedures were introduced in the middle of the session, and the goats resumed the sessions without the correction procedure after they passed the performance criterion using the correction procedure. (**B**) Stage 2 performances. (**C**) Stage 3 performances.

**Table 1 animals-15-02115-t001:** The information of the goat participants.

Name	Sex	Age (Years Old)	Kinship	Participated in
Hikari	Female	Older than 9	-	Pilot, Stage 1, 2, 3
Yoshino	Female	8	Nishiki’s daughter; Ryu’s sibling	Pilot, Stage 1
Nishiki	Female	11	Yoshino and Ryu’s mother	Pilot
Ryu	Male	8	Nishiki’s son; Yoshino’s sibling	Pilot, Stage 1, 2

**Table 2 animals-15-02115-t002:** Performance and the binomial test results compared to the chance level of the pilot test (*: *p* < 0.05; **: *p* < 0.01; ***: *p* < 0.001).

	Distractor Number	1		2		3		4		5	
Participant		Accuracy	*p*-Value	Accuracy	*p*-Value	Accuracy	*p*-Value	Accuracy	*p*-Value	Accuracy	*p*-Value
Hikari		62.5%	0.076	62.5%	<0.001 ***	75.0%	<0.001 ***	66.7%	<0.001 ***	41.7%	<0.001 ***
Yoshino		58.3%	0.154	45.8%	0.068	41.7%	0.021 *	58.3%	<0.001 ***	50.0%	<0.001 ***
Ryu		50.0%	0.419	50.0%	0.028 *	62.5%	<0.001 ***	54.2%	<0.001 ***	58.3%	<0.001 ***
Nishiki		70.8%	0.011 *	58.3%	0.003 **	41.7%	0.021 *	33.3%	0.036 *	20.8%	0.200

**Table 3 animals-15-02115-t003:** Performance and the binomial test results compared to the chance level of the odd-item search training (***: *p* < 0.001).

	Stage 1						Stage 2						Stage 3					
	Overall ^†^		First Four Sessions		Last Four Sessions		Overall		First Four Sessions		Last Four Sessions		Overall		First Four Sessions		Last Four Sessions	
Participant	Accuracy	*p*-Value	Accuracy	*p*-Value	Accuracy	*p*-Value	Accuracy	*p*-Value	Accuracy	*p*-Value	Accuracy	*p*-Value	Accuracy	*p*-Value	Accuracy	*p*-Value	Accuracy	*p*-Value
Hikari	69.7%	<0.001 ***	58.3%	0.097	81.3%	<0.001 ***	66.3%	<0.001 ***	56.3%	0.157	77.1%	<0.001 ***	51.7%	0.326	47.9%	0.557	52.1%	0.443
Ryu	68.5%	<0.001 ***	45.8%	0.333	85.4%	<0.001 ***	24.2%	<0.001 ***	18.8%	<0.001 ***	37.5%	0.056	-	-	-	-	-	-
Yoshino	11.7%	<0.001 ***	10.4%	<0.001 ***	5.38%	<0.001 ***	-	-	-	-	-	-	-	-	-	-	-	-

† Excluding the sessions with correction procedures.

## Data Availability

All data are available in the manuscript and Appendix A.

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
