# Peer review of "Touchscreen Tasks for Cognitive Testing in Domestic Goats (Capra hircus): A Pilot Study Using Odd-Item Search Training"

_animals, 2025, doi:10.3390/ani15142115_

Round 1
Reviewer 1 Report
Comments and Suggestions for Authors
The following may be taken care of before the manuscript is accepted:
1. The authors are strongly advised to use third person while describing the experiment. At majority of the places, first person has been used.
2. The authors have used, terminologies for describing the experiment- condition, stage, session and trial. However, these terminologies have not been described. So it becomes difficult for the reader to grasp the essence of variety of set-ups. Kindly ensure that the basic design of experiment is made amply clear to the reader. Methods portion needs to be revised extensively.
3. The authors may also clarify that how it is ensured that correction procedure during odd-item search training does not lead to learning.
4. The number of sessions and number of trainings for each goat is variable. With each session and training, pattern learning occurs. So this needs to be taken into account while interpreting the results.

The authors have repeatedly used first person in text. It is strongly advised that third person may be used to describe events, and observations. Some sentences lack clarity, which have been marked in manuscript. May be improved.
Reviewer 2 Report
Comments and Suggestions for Authors
There are some interesting aspects to this paper and the authors are to be commended for implementing a training program that demonstrated aspects of goats' visual discrimination abilities. The MS requires considerable revision however.
Some general comments:
There is a considerable literature exploring the visual discrimination abilities of farmed species and horses-some of which you did cite. However you did not engage with this in any depth- that is, compare and contrast what was found. This was particularly the case in the discussion where it is essential to compare your work with that of others who have researched in the same field.
While you did successfully train goats to use a touchscreen (TS), which is noteworthy, far too much of the MS was spent discussing this, rather than the evidence of the learning that the experiment tested. The sections on the TS, particularly in the discussion could be considerably shortened and a greater focus on what the animals learned and the likely underlying learning processes could be undertaken.
The methods description was hard to follow and which hindered my understanding of the results.
There is insufficient information about what statistical tests were actually used-this must be added.
Lastly, there was an over-reliance on descriptive text in regards to the goats' learning performance and insufficient discussion of the numerical data that was recorded.
I have made extensive notes on the attached MS in regards to the above points and other issues that require attention.
The SM video benefits the understanding of the method, however it would be beneficial to include footage of the later stages as well to demonstrate how the more complex use of the pictograms was implemented as it is not clear from the methods description.

The tone of the writing could be improved as, currently it is quite informal, especially with when referring to the goats.
Round 2
Reviewer 1 Report
Comments and Suggestions for Authors
The authors have greatly improved the language and explanation. No change is required in MS with respect to manuscript language. Majority of the queries raised have been answered, However, authors have not been able to still explain, how they can explain pattern learning while training/ correction procedure. They have cited it as one of the shortcomings of this work, but explanation is required.
Following may kindly be improved upon:
1. The introduction portion may be shortened. Line 124 to 129, is not required in introduction part.
2. Age has been cited as one of the possible reasons of poor learning exhibited by one of the goat (No. of animals is very less to draw this conclusion). Can sex difference be another reason? Since, Ryu being male is exhibiting better learning and discriminatory capabilities, than Hikari and Yoshino (Both being females) during pilot testing and Hikari during Odd-even search training.
2. Number of sessions (during both touch screen training and odd item search training) as well as correction procedure, for each goat is different. Authors may explain difference in results exhibited by each goat with respect to number of sessions.

Reviewer 2 Report
Comments and Suggestions for Authors
The authors are to be commended for addressing my comments in detail. The manuscript is significantly improved. The supplementary videos should include the original early training video and the new video which demonstrates the distractor trials as this considerably contributes to the methods description.
